# The Workability and Mechanical Performance of Fly Ash Cenosphere–Desert Sand Ceramsite Concrete: An Experimental Study and Analysis

**DOI:** 10.3390/ma16031298

**Published:** 2023-02-02

**Authors:** Junlin Guo, Kang Yuan, Jianjiang Xu, Ying Wang, Dan Gan, Mingsheng He

**Affiliations:** 1College of Water Conservancy & Architectural Engineering, Shihezi University, Shihezi 832003, China; 2School of Civil Engineering, Chongqing University, Chongqing 400045, China

**Keywords:** lightweight aggregate concrete, ceramsite concrete, desert sand, fly ash cenosphere, polymer emulsion, workability and mechanical performance, orthogonal test

## Abstract

In order to alleviate the shortage of sand resources for construction, make full use of industrial waste and promote the development of green lightweight aggregate concrete in the desert and surrounding areas, this paper proposes a new lightweight ceramsite concrete, fly ash cenospheres and desert sand ceramsite concrete (FDCC). An orthogonal test was conducted to analyze the effects of the desert sand (DS) replacing ratio, fly ash cenosphere (FAC) replacing ratio and polymer emulsion (PLE) addition on the damage patterns, slump, apparent density and compressive strength of the FDCC. The results showed that the most influential factors for the slump, apparent density and compressive strength of the FDCC were the FAC replacing ratio, FAC replacing ratio and DS replacing ratio, respectively. Meanwhile, the PLE addition had little effect on the workability or mechanical performance of the FDCC. With the increase in the DS replacing ratio, the slump decreased rapidly and the compressive strength reached its peak value, increasing by 20.6% when the DS replacing ratio was 20%. With the increase in the FAC replacing ratio, the slump increased by 106%, the apparent density decreased gradually and the compressive decreased and then increased, reaching its lowest value when the FAC replacing ratio was 20%. According to the synthetic evaluation analysis, the optimum DS replacing ratio, FAC replacing ratio and PLE addition of the FDCC were 20%, 30% and 1%, respectively.

## 1. Introduction

Lightweight aggregate concrete (LWAC) has been widely used in high-rise buildings, large-span bridges and offshore platforms due to its various advantages, including its light weight, strength, good seismic performance, good economic efficiency, good durability and good heat insulation performance [1,2,3,4,5,6,7]. Ceramsite concrete is a type of LWAC that is relatively more widely used as a structural concrete than pumice, volcanic, cinders, expanded clay or other lightweight aggregates because of its wide range of raw materials and its good seismic performance [8,9]. As river sand (RS) is the most widely used fine aggregate in ordinary and lightweight aggregate concrete, it has been over-exploited despite its non-renewability, which has caused serious damage to the ecological environment and increased the prices of concrete [10,11,12]. Therefore, finding reasonable RS substitutes and utilizing them to prepare green lightweight ceramsite concrete could help to alleviate the shortage of construction sand, protect the environment and promote the development of green LWAC.

Desert sand (DS) is considered to be an appropriate alternative to RS. The utilization of DS could reduce RS consumption and improve waste utilization, environmental protection and material localization in deserts and surrounding areas [13]. Jayawardena [7] prepared C25 and C30 DS concrete, proving that DS could be used to prepare concrete directly, without any special treatment. Zhou [14] investigated the effects of the DS replacing ratio, fly ash mixing amount and sand rate on the compressive strength of concrete under low temperatures. The results showed that DS concrete demonstrated good mechanical performance at low temperatures. Li [15] studied the mechanical properties of ordinary concrete with a large DS replacing ratio and the results indicated that the compressive strength, splitting tensile strength and workability of DS concrete were sufficient when the DS addition was 62% and 80%. Park [16] found that the rheological properties of DS concrete increased with the increase in the DS mixing amount. Xue [17] revealed the good chlorine resistance of DS concrete using a super-depth 3D microscope, X-ray diffraction (XRD) and nuclear magnetic resonance (NMR). Li [18] investigated the seismic performance of RC columns that were made from DS concrete and found that having a DS replacing ratio that was in the range of 20–40% moderately improved the seismic behavior of the DS-based RC columns. Therefore, according to the literature, the utilization of DS as a fine aggregate instead of RS could meet the requirements for the working and mechanical properties of concrete.

However, using DS instead of RS in equal volumes would increase the bulk density of concrete. Fly ash cenospheres (FACs) are by-products from fly ash that are spherical, hollow and lightweight and have good heat resistance [19,20]. FACs could be used to reduce the bulk density of DS concrete, and they could also be beneficial for industrial waste utilization and environmental protection. Hanif [21] found that FACs could be used as a construction material for load-bearing elements and that the optimum dosage was approximately 50%. Patel [22] used FACs to prepare concrete with a strength of 20–35 MPa. Zhang [23] revealed that the compressive and flexural strength of modified FAC concrete increased with the FAC addition and that the microstructure was stronger than unmodified FAC concrete.

Ceramsite concrete is brittle due to its aggregate properties; for example, the polymer emulsion (PLE) can dehydrate during the hardening process of concrete and form a spatial mesh structure [24,25]. Therefore, adding PLE to ceramsite concrete can improve its brittleness, durability and mechanical properties, as well as improving the microstructure of the concrete.

In general, according to our literature review, DS and FACs could both individually replace RS as the fine aggregate in concrete. However, research on the use of these two materials together to prepare LWAC has been not found, and the influence of the DS and FAC replacing ratios on workability and mechanical performance is not clear. Therefore, we proposed a new lightweight ceramsite concrete that used fly ash cenospheres (FACs) and desert sand (DS) to replace river sand as the fine aggregate, namely fly ash cenosphere–desert sand ceramsite concrete (FDCC). An orthogonal test of three factors and four levels was conducted, as well as a microstructural analysis of the FDCC using scanning electron microscopy (SEM). The effects of different DS replacing ratios, FAC replacing ratios and polymer emulsion (PLE) additions on the damage patterns, slump, apparent density and compressive strength of the FDCC were investigated, and reasonable DS and FAC replacing ratios and PLE additions were determined. The results of this study are expected to provide guidelines and a reference for the use of DS and FACs in engineering construction in deserts and surrounding areas.

## 2. Materials and Methods

### 2.1. Materials

In this study, the binding materials were ordinary Portland cement and fly ash, as shown in Figure 1a,b. The fine aggregates were RS, DS and FACs, as shown in Figure 1c–e. The particle size, bulk density and apparent density of the RS were less than or equal to 5 mm, 1420 kg/m^3^ and 2239 kg/m^3^, respectively. The DS was from the south edge of the Gurbantünggüt Desert in China, with an average particle size of 0.183 mm and a bulk density and apparent density of 1610 kg/m^3^ and 2690 kg/m^3^, respectively. The particle size distributions of the DS and RS are shown in Figure 2.

The particle size, bulk density and apparent density of the FACs were 1 mm–2 mm, 303 kg/m^3^ and 1877 kg/m^3^, respectively. The chemical compositions of the binding materials and fine aggregates are shown in Table 1. The coarse aggregate was shale ceramsite, as shown in Figure 1f, which had a particle size of 10 mm–20 mm and a bulk density and apparent density of 559 kg/m^3^ and 1348 kg/m^3^, respectively. The other physical properties are shown in Table 2. In addition, PLE and a water reducing agent were used as the additives. The PLE was a benzine emulsion with a solid content of 48–50%, while the water reducing agent was a polycarboxylic acid water reducing agent with a water reducing rate of 25%. The main chemical properties of the water reducing agent are shown in Table 3. The mixing water was tap water.

### 2.2. Orthogonal Test Design

Orthogonal testing can effectively reduce the required specimen amounts and test times. They are generally conducted by selecting typical points from full trials using orthogonality [26,27,28]. An orthogonal test scheme with three factors and four levels was adopted in this study. The three factors were the DS replacing ratio, FAC replacing ratio and PLE addition, which were denoted as ***A***, ***B*** and ***C***, respectively. The four levels of the DS and FAC replacing ratios were 0, 10%, 20% and 30%, while the four levels of PLE addition were 0, 0.5%, 1% and 1.5%, as shown in Table 4. The last column in Table 4 is empty because of a factor denoted as ***D***, which was used for error verification. It should be noted that the DS and FAC replacing ratios were the volume fractions of the fine aggregates, and the PLE addition was the mass fraction of solid content in the PLE to the binding materials. The orthogonal test was designed according to the selected factors and levels, and the L16(4^4^) orthogonal table is shown in Table 5.

The loose volume method was adopted to design the reference mix ratio according to the following Chinese code: Technical specifications for the application of lightweight aggregate concrete (JGJ/T12-2019) [29]. The water–binding material ratio was 0.35, the binding material mass was 460 kg/m^3^, fly ash comprised 20% of the total binding material mass, the total volume of the coarse and fine aggregate was 1.3 m^3^, the sand rate was 35% and the water reducing agent dosage was 1.2% of the binding material. The mix ratios for all test conditions are shown in Table 6. 

### 2.3. Specimen Preparation

To prevent the high water absorption of ceramsite from affecting the mixing effect of the concrete, the ceramsite was pre-wetted for 2 h. The specimens were mixed and vibrated artificially. Firstly, the cement, fly ash, DS, RS and FACs were premixed together for 2 min. Then, the pre-wetted ceramsite was added and the mixture was mixed continuously for 2 min. Finally, water and the water reducing agent and PLE were added and the mixture was mixed for another 2 min. After completing the slump test, the fresh concrete was cast into six cubic molds, which were 100 mm × 100 mm × 100 mm in size. The specimens were demolded and remarked after 24 h and they were then cured in a standard curing box with humidity greater than 95% and a curing temperature of 20 ± 2 ℃. Of the six cubes, three were cured for 7 days and three were cured for 28 days. The specimen preparation and curing processes are shown in Figure 3.

### 2.4. Test Methods 

#### 2.4.1. Slump

The slump of the FDCC was measured according to the following Chinese code: Standard for the test method for the performance of ordinary fresh concrete (GB/T 50080–2016) [30]. The fresh concrete was loaded into the slump barrel in three even layers. Each layer of concrete was pounded 25 times from the edges to the center with a vibrating bar. After pounding the top layer, excess concrete was scraped off and the concrete was smoothed along the opening of the barrel. Then, the slump barrel was lifted vertically and smoothly and placed near the specimens. When the specimens stopped slumping or the slumping time was greater than 30 s, the distance between the top of the slump barrel and the top of the slumped concrete specimen was measured as the slump value of the concrete specimen. The slump test device is shown in Figure 4a.

#### 2.4.2. Compressive Strength and Apparent Density

The compressive strength test was carried out according to the following Chinese code: Standard for the test method for the mechanical properties of ordinary concrete (GBT 50081–2019) [31]. A 600-kN electronic universal testing machine was selected as the loading device, as shown in Figure 4b. The compressive force values were collected by the data acquisition system of an electronic universal testing machine. The concrete specimens were pre-compressed before the test and the loading rate was 0.5 MPa/s. The apparent density of the concrete specimens was obtained by dividing the mass of the specimen by its natural volume. 

#### 2.4.3. Microstructural Analysis 

After the compressive strength test, the damaged specimens were immersed in anhydrous ethanol to stop the hydration of the material. Then, they were dried in an oven at a temperature of 60 °C. An SEM (Regulus8100, made in HITACHI, Tokyo, Japan) was selected to scan the microstructure of the FDCC, as shown in Figure 4c. Before the test, the samples were made into square blocks that were no more than 5 mm in size and covered with a gold coating to enhance their electrical conductivity.

## 3. Results and Analysis

### 3.1. Test Results

The results of the slump, apparent density and 7-d and 28-d cubic compressive strength tests are shown in Table 7. The apparent density of the FDCC-16 specimen was the lowest, with a value of 1695.7 kg/m^3^, while the FDCC-13 specimen had the greatest apparent density, with a value of 1849.6 kg/m^3^. The 28-d cubic compressive strength of the FDCC-9 specimen was the greatest, with a value of 32.72 MPa, which was 6.4% higher than that of the FDCC-1 specimen.

### 3.2. Damage Patterns

The compressive damage patterns of the FDCC are shown in Figure 4. Diagonal cracks appeared on the surfaces of the specimens, which then extended toward the center during the compressive strength test. The damage patterns of the FDCC presented typical "X" shapes, as shown in Figure 5a. The lightweight aggregate was split and damaged, while the ceramsite particles were distributed uniformly on the damaged surfaces. No uneven distribution phenomena that were caused by floating ceramsite particles occurred in the aggregates, as shown in Figure 5b. Compared to ordinary LWAC, the addition of FACs reduced the strength of the concrete cement mortar. Therefore, splitting damage to the FACs occurred in the FDCC, as shown in Figure 5c.

### 3.3. Range and Variance Analysis

The range analysis method can increase the test efficiency by distributing multi-factor and multi-level tests evenly over a smaller number of tests [32]. Then, the primary and secondary sequences of factors can be determined though a range analysis. The data for the range and variance analysis of the slump, apparent density and cubic compressive strength of the FDCC are presented in Table 7, considering the factors of DS replacing ratio, FAC replacing ratio and PLE addition. The range analysis results are presented in Table 8. 

In Table 8, $K_{ij}$ is the test result of the ith factor under the influence of the jth level. The calculation formula for this is shown in Equation (1), where $R_{i}$ is the difference between the maximum and minimum values of the ith factor under the influence of the jth level. The calculation formula for this is shown in Equation (2).
(1)$$K_{ij}=\frac{\sum_{m=1}^{n}B_{ij,m}}{n}$$
(2)$$R_{i}=\max\left\{K_{ij}\right\}-\min\left\{K_{ij}\right\}$$
where m is the specimen; i and j are the factor and level, respectively; $B_{ij,m}$ is the test result of the ith factor of the mth specimen under the influence of the jth level, and n is the number of calculated results for the ith factor under the influence of the jth level. 

The magnitude of the influence of each factor on the slump of the FDCC was as follows: FAC replacing ratio *(**B***) > DS replacing ratio (***A***) > PLE addition (***C***). The magnitude of the influence of each factor on the apparent density of the FDCC was as follows: FAC replacing ratio (***B***) > DS replacing ratio (***A***) > PLE addition (***C***). The magnitude of the influence of each factor on the 7-d and 28-d compressive strength of the FDCC was as follows: DS replacing ratio (***A***) > FAC replacing ratio (***B***) > PLE addition (***C***). 

The range analysis method is intuitive and simple, but variations in and the randomness of levels can lead to errors in test results, while the variance analysis method can increase the precision of test results and reduce errors [33]. Our variance analysis is shown in Table 9, where F represents the test value of the variance analysis, and F_0.01_(3,3), F_0.05_(3,3) and F_0.1_(3,3) represent the critical values of F that were used to evaluate the different degrees of influence of the factors on the performance of the FDCC. When F > F_0.01_, the influence of the factor was highly significant (denoted as ***). When F > F_0.1_, the influence of the factor was affected (denoted as *). 

As can be seen from Table 9, the DS and FAC replacing ratios were the main factors that affected the slump of the FDCC. The effect of the FAC replacing ratio on the apparent density of the FDCC was affected, whereas the DS replacing ratio was the main factor that affected the compressive strength of the FDCC. The primary and secondary sequences of the factors affecting the slump, apparent density and cubic compressive strength of the FDCC could be derived from the variance analysis. The results were consistent with those of the range analysis.

### 3.4. Parametric Analysis

#### 3.4.1. DS Replacing Ratio

Figure 6 shows the relationship between the DS replacing ratio and the properties of the FDCC. As can be seen from the figure, the slump decreased by 49.48% when the DS replacing ratio increased from 0 to 30%. This decreasing trend was the fastest when the DS replacing ratio increased from 10% to 20%. The main reason for this was the DS particle size distribution: the DS was very fine sand with a high specific surface area [34], which decreased the fluidity of the FDCC. The apparent density of the FDCC increased and then decreased with the increase in the DS replacing ratio. The apparent density of the FDCC was 4.01% higher than that of ordinary ceramsite concrete when the DS replacing ratio was 20%.

The 7-d and 28-d compressive strength increased and then decreased with the increase in the DS replacing ratio, reaching their peak values with DS replacing ratios of 20%. This could have been due to the fact that the very fine DS particles filled the gaps between the aggregates and enhanced the compactness of the cement mortar. Moreover, the bonding effect between the aggregates and the binding materials improved because of the pozzolanic effect of the DS [8], thereby increasing the compressive strength of the FDCC. However, when the DS replacing ratio reached a certain level, there was not enough cement slurry to cover all of the fine aggregate particles, which resulted in a decrease in compressive strength.

Therefore, considering the effect of the DS replacing ratio on the slump, apparent density and compressive strength of the FDCC comprehensively, the most suitable DS replacing ratio was determined to be 20%.

#### 3.4.2. FAC Replacing Ratio

Figure 7 shows the relationship between the FAC replacing ratio and the properties of the FDCC. The slump of the FDCC increased with the FAC replacing ratio and the slump of the FDCC was 106.8% higher than that of ordinary ceramsite concrete when the FAC replacing ratio was 30%. Meanwhile, the apparent density of the FDCC decreased by 4.44%. This was because the spherical morphology of the FACs improved the fluidity and reduced the adhesiveness of the concrete [35,36], which reduced the friction between the cement mortar and the coarse aggregate, thus improving the workability of the concrete [8].

The compressive strength of the FDCC decreased and then increased with the increase in the FAC replacing ratio. The 7-d and 28-d compressive strengths of the FDCC were 15.71% and 15.19% lower than those of ordinary ceramsite concrete, respectively, when the FAC replacing ratio was 20%, and they were 8.82% and 5.38% higher, respectively, when the FAC replacing ratio increased from 20% to 30%. Because of the relatively low strength of FACs, adopting FACs as a replacement for RS decreased the strength of the cement mortar, which led to a decrease in the strength of the FDCC. However, the FAC surface was active, which promoted the hydration reaction and enhanced the adhesion of the FAC interface. This could have been the reason for the increase in the compressive strength of the FDCC.

Therefore, considering the effect of the FAC replacing ratio on the slump, apparent density and compressive strength of the FDCC comprehensively, the most suitable FAC replacing ratio was determined to be 30%.

#### 3.4.3. PLE Addition

Figure 8 shows the relationship between the PLE addition and the properties of the FDCC. As can be seen in the figure, the slump of the FDCC decreased and then increased with the increase in the PLE addition, reaching the minimum value when the PLE addition was 0.5%. The apparent density of the FDCC slightly increased with the PLE addition and was 2.03% higher than that of ordinary LWAC when the PLE addition was 1.5%. The PLE was viscous, which caused the slump of the FDCC to decrease; however, due to the small amounts that were added, the effect of the PLE addition on the apparent density of the FDCC was generally insignificant.

The compressive strength of the FDCC increased and then decreased with the increase in the PLE addition, reaching a maximum value of 29.2 MPa when the PLE addition was 1%. This could have been due to the viscosity of the PLE addition increasing the bond strength between the binding materials and the aggregates, which led to the increase in the compressive strength of the FDCC. However, with the increase in the PLE addition, more polymer developed between the binding material and the aggregates, which disrupted the production of hydration products and reduced the bonding action between the binding materials and the aggregates, resulting in a decrease in the compressive strength of the FDCC.

Therefore, considering the effect of the PLE addition on the slump, apparent density and compressive strength of the FDCC comprehensively, the most suitable PLE addition was determined to be 1%.

## 3.5. SEM Analysis

The microstructures of the DS and FACs, as determined by the SEM, are shown in Figure 9. It can be seen from the figure that the DS was irregular and had a relatively smooth surface. The FACs were spherical and had a few pores and bumps on their outer surfaces. In addition, the microstructures of the damaged surfaces of the specimens and the transition zones between the aggregates and the matrix were observed to analyze the compactness of the interfaces and the distributions of the aggregates. The corresponding results are shown in Figure 10 and Figure 11.

Figure 10 shows some damaged sections of the FDCC. FACs and DS particles can be observed on the scanning surface in Figure 10a. The DS particles were fine and filled the gaps between the aggregates, as shown in Figure 10b, which could explain the phenomenon of the apparent density and compressive strength of the FDCC increasing when the DS replacing ratio reached a certain level. However, slight cracks appeared at the edges of the DS particles (Figure 10b), which formed a weak position and became damaged under mechanical action because the DS surface was relatively smooth, meaning that damage occurred more easily at the interfaces [37].

Splitting damage to the FAC particles is shown in Figure 10a. Some hydration products were observed on the inner walls of the FACs but there were no cracks at the outer interfaces of the FACs, as shown in Figure 10c. This illustrated that some water entered the inner portions of the FACs though the pores in their walls and that a hydration reaction occurred and increased the bonding properties of the binding material and the FACs. Meanwhile, the strength of the FAC was still lower than that of the binding material and other aggregates, which generally caused the strength of the FDCC to decrease when the FAC replacing ratio increased, agreeing with our previous conclusions.

There were no cracks at the interfaces between the binding material and the ceramsite, as shown in Figure 10d. This was due to the fact that the pre-wetting of the ceramsite prevented it from absorbing water from the binding material slurry and reduced the shrinkage of the binding material slurry during the concrete hardening process.

Figure 11 shows the microstructures of the FDCC with DS replacing ratios of 0%, 10%, 20% and 30%. Compactness is beneficial for the strength of FDCC, while the microstructure of ordinary LWAC is not compact. The main hydration products were CH and CSH, as shown in Figure 11a. The microstructure of the FDCC became more compact as the DS replacing ratio increased, which showed that the aggregates were covered well. However, there was not much difference in the compactness of the FDCC between the DS replacing ratios of 20% and 30%. The very fine DS particles demonstrated heterogeneous nucleation and a pozzolanic effect [9], which promoted the hydration reaction, increased the compactness of the FDCC and improved the strength of the FDCC. However, the compressive strength of the FDCC decreased when the DS replacing ratio decreased from 20% to 30%, which could have been because the binding material could not cover all of the DS particles, meaning that cracks occurred, which then decreased the mortar strength of the FDCC.

## 4. Conclusions

In order to alleviate the shortage of sand resources for construction, make full use of industrial waste and promote the development of green lightweight aggregate concrete in the desert and surrounding areas, we proposed the adoption of DS and FACs to replace RS when preparing a new lightweight ceramsite concrete (FDCC). The workability and mechanical performance of the FDCC were investigated and analyzed using an orthogonal test and SEM. The following conclusions were drawn.

(1)The three factors of the DS replacing ratio, FAC replacing ratio and PLE addition had different effects on the workability and mechanical performance of the FDCC. According to the test results, the FAC replacing ratio was the most influential factor in the slump and apparent density of the FDCC, followed by the DS replacing ratio and PLE addition. For compressive strength, the most influential factor was the DS replacing ratio, followed by the FAC replacing ratio and PLE addition. The PLE addition had little effect on the workability or mechanical performance of the FDCC.(2)With the increase in the DS replacing ratio from 0 to 30%, the slump decreased from 72.8 mm to 36.8 mm, the apparent density increased gradually and the compressive strength increased by 20.6% to reach its peak value when the DS replacing ratio was 20% and then it decreased. With the increase in the FAC replacing ratio from 0 to 30%, the slump increased by 106% from 36.65 mm to 76 mm, the apparent density decreased gradually and the compressive strength first decreased and then increased, reaching its lowest value when the FAC replacing ratio was 20%.(3)According to our synthetic evaluation analysis of the orthogonal test results, the optimum DS replacing ratio, FAC replacing ratio and PLE addition for the FDCC were 20%, 30% and 1%, respectively. DS and FACs could replace up to 50% of the river sand in the FDCC, which means that full use can be made of desert sand and industry waste, thereby reducing the demand for river sand. This would also be beneficial for the environment.(4)In this study, we found that the lower strength of desert sand and fly ash cenospheres as fine aggregates increased the brittleness of the FDCC. Therefore, some types of fibers could be used in FDCC to improve its toughness and strength. Additionally, the long-term properties of FDCC may be worth paying attention to in the future.

## Figures and Tables

**Figure 1 materials-16-01298-f001:**
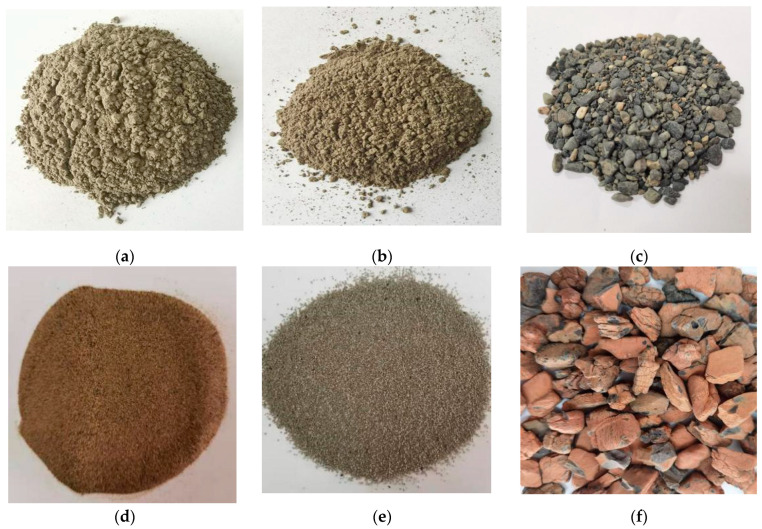
The raw materials: (**a**) cement; (**b**) fly ash; (**c**) RS; (**d**) DS; (**e**) FACs; (**f**) shale ceramsite.

**Figure 2 materials-16-01298-f002:**
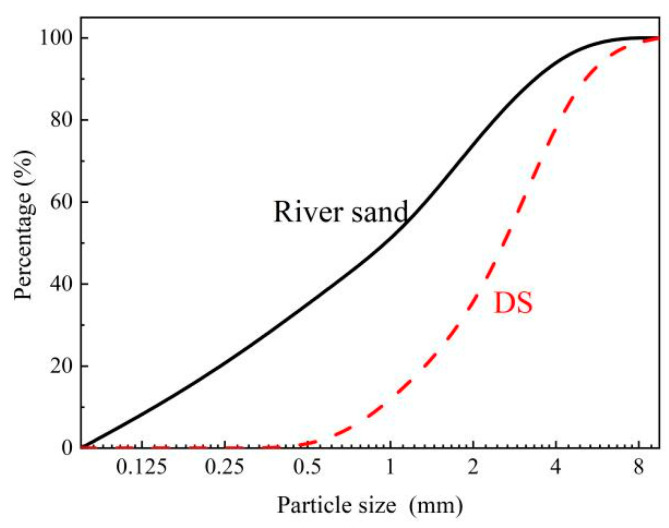
The particle size distribution curve.

**Figure 3 materials-16-01298-f003:**
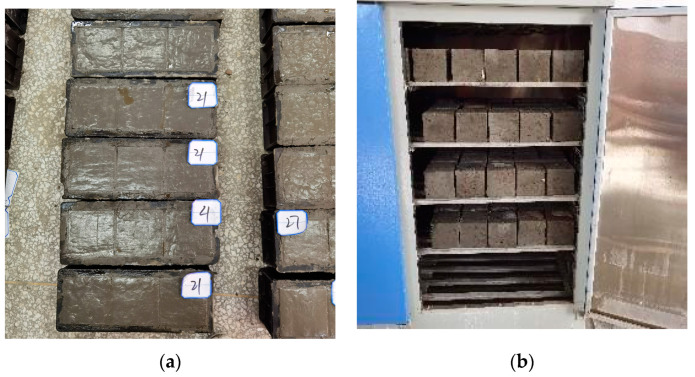
The specimens’ preparation and curing: (**a**) specimen preparation; (**b**) specimen curing.

**Figure 4 materials-16-01298-f004:**
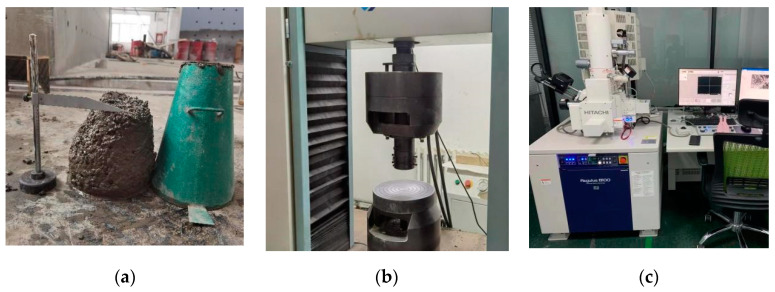
The test devices: (**a**) slump; (**b**) compressive strength; (**c**) SEM.

**Figure 5 materials-16-01298-f005:**
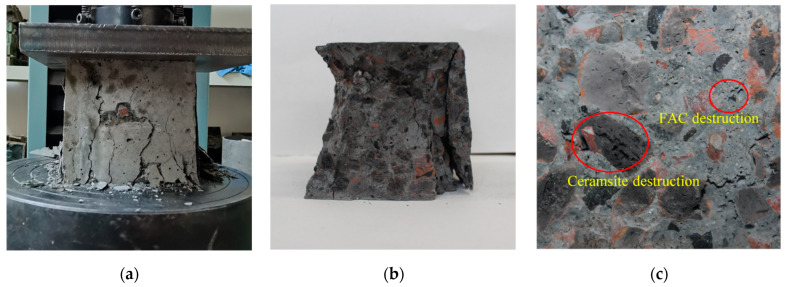
The damage patterns of the FDCC: (**a**) surface cracks; (**b**) overall damage; (**c**) aggregate damage.

**Figure 6 materials-16-01298-f006:**
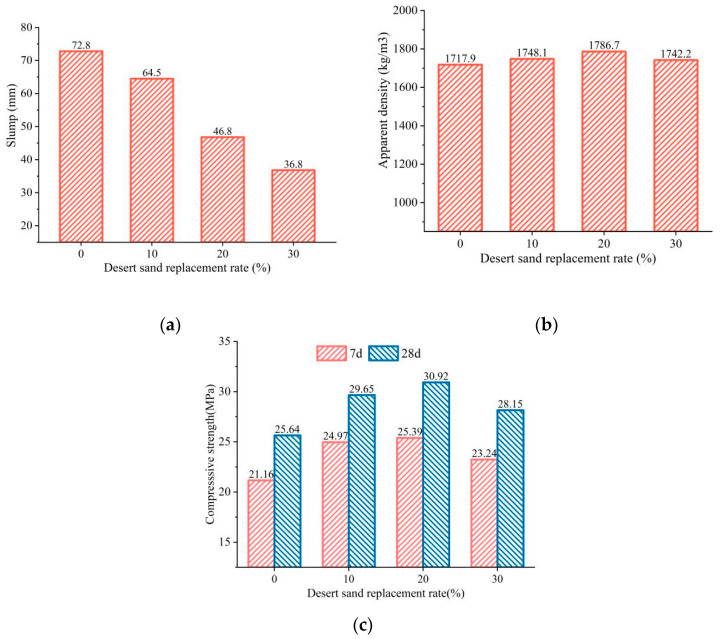
The relationship between the DS replacing ratio and the FDCC performance: (**a**) slump; (**b**) apparent density; (**c**) compressive strength.

**Figure 7 materials-16-01298-f007:**
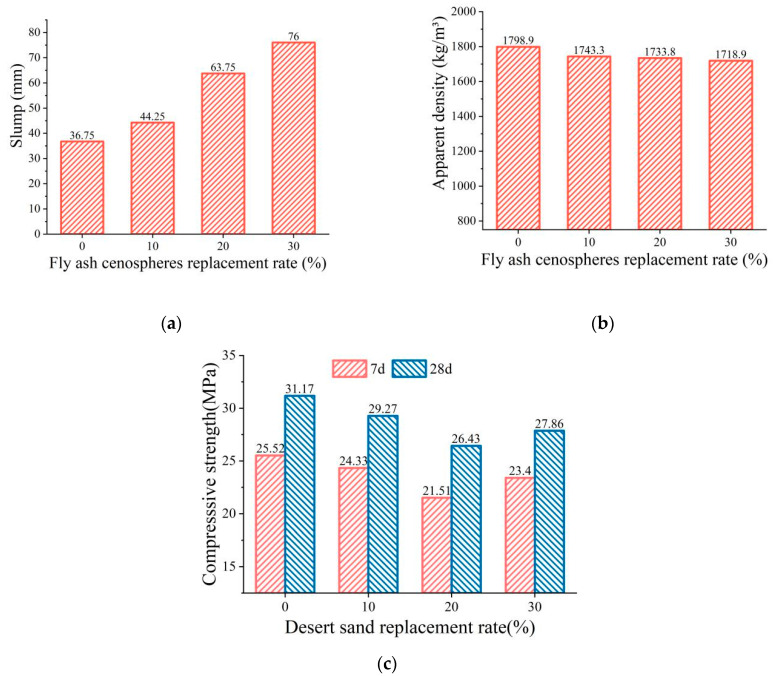
The relationship between the FAC replacing ratio and the FDCC performance: (**a**) slump; (**b**) apparent density; (**c**) compressive strength.

**Figure 8 materials-16-01298-f008:**
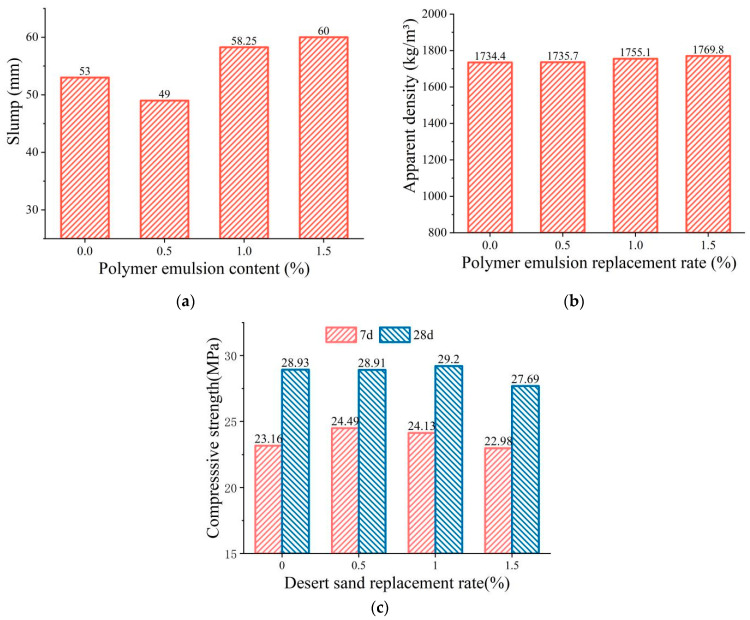
The relationship curve between PLE addition and FDCC performance: (**a**) slump; (**b**) apparent density; (**c**) compressive strength.

**Figure 9 materials-16-01298-f009:**
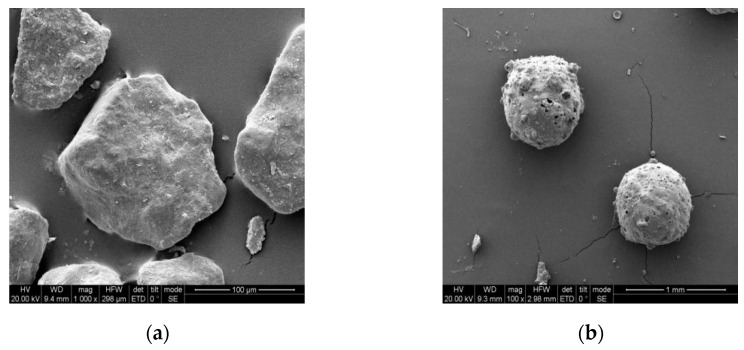
SEM of DS and FAC: (**a**) DS; (**b**) FAC.

**Figure 10 materials-16-01298-f010:**
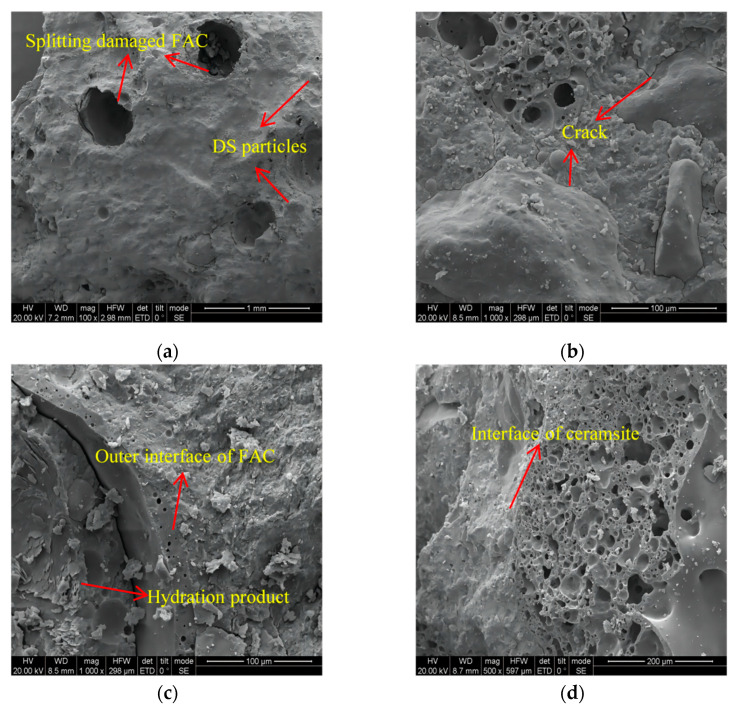
**The** SEM images of the FDCC aggregate interfaces: (**a**) a damaged specimen; (**b**) a DS interface; (**c**) an FAC interface; (**d**) a ceramsite interface.

**Figure 11 materials-16-01298-f011:**
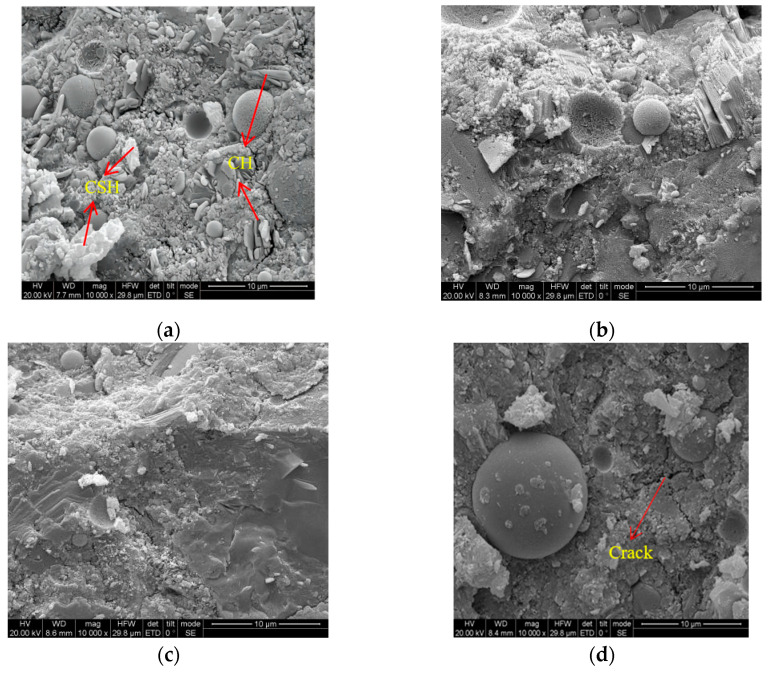
The SEM images of the FDCC under different DS replacing ratios: (**a**) 10%; (**b**) 20%; (**c**) 30%; (**d**) 40%.

**Table 1 materials-16-01298-t001:** The chemical compositions of the raw materials.

Composition (wt.%)	SiO_2_	Al_2_O_3_	Fe_2_O_3_	K_2_O	CaO	SO_3_	MgO	Na_2_O	Others
Cement	18.69	3.95	4.32	0.62	65.26	3.72	1.52	0.83	1.09
Fly ash	45.68	16.72	10.42	2.10	13.37	1.73	4.37	3.81	1.8
DS	64.58	9.48	2.32	1.97	8.62	1.09	2.06	2.43	7.45
FAC	59.44	23.3	5.49	2.85	2.31	1.52	1.37	1.49	2.23
RS	90.98	3.42	0.56	1.96	1.01	0.63	0.20	0.26	0.98

**Table 2 materials-16-01298-t002:** The performance indices of the shale ceramsite.

Technical Index	Bulk Density (kg/m3)	Numerical Tube Pressure (MPa)	Water Absorption (%)	Softening Coefficient	Silt Content (%)	Mass Loss After Boiling (%)
Standard	-	≥1.0	≤10	≥0.8	≤3.0	≤5.0
Actual value	559	3.6	4.3	1	0.2	1

**Table 3 materials-16-01298-t003:** The chemical properties of the water reducing agent.

Main Component (%)	Salt	Isopropanol	Acetone	Sodium Citrate	Sodium Hydroxide	Pigment
Water reducing agent	28.71	0.69	2.14	4.51	2.89	0.06

**Table 4 materials-16-01298-t004:** The factor levels in the orthogonal test.

Level	Factor			
	*A*(%)	*B* (%)	*C* (%)	*D*
1	0	0	0	1
2	10	10	0.5	2
3	20	20	1	3
4	30	30	1.5	4

**Table 5 materials-16-01298-t005:** The orthogonal test.

Test Number	Factor			
	*A*	*B*	*C*	*D*
FDCC1	0	0	0	1
FDCC2	0	10	0.5	2
FDCC3	0	20	1	3
FDCC4	0	30	1.5	4
FDCC5	10	0	0.5	3
FDCC6	10	10	0	4
FDCC7	10	20	1.5	1
FDCC8	10	30	1	2
FDCC9	20	0	1	4
FDCC10	20	10	1.5	3
FDCC11	20	20	0	2
FDCC12	20	30	0.5	1
FDCC13	30	0	1.5	2
FDCC14	30	10	1	1
FDCC15	30	20	0.5	4
FDCC16	30	30	0	3

**Table 6 materials-16-01298-t006:** The dosage of each material.

Test Number	Amount of Raw Material (kg/m^3^)								
	Water	Cement	Fly Ash	Ceramsite	RS	DS	FAC	PLE	Water Reducing Agent
FDCC1	154.0	352.0	88	559	646.1	0	0	0	4.6
FDCC2	151.6	349.8			581.4	0	15.9	4.5	
FDCC3	149.2	347.6			516.8	0	31.8	9.1	
FDCC4	146.8	345.4			452.2	0	47.7	13.7	
FDCC5	151.6	349.8			581.4	73.4	0	4.5	
FDCC6	154.0	352.0			516.8	73.4	15.9	0	
FDCC7	146.8	345.4			452.2	73.4	31.8	13.7	
FDCC8	149.2	347.6			387.6	73.4	47.7	9.1	
FDCC9	149.2	347.6			516.8	146.9	0	9.1	
FDCC10	146.8	345.4			452.2	146.9	15.9	13.7	
FDCC11	154.0	352.0			387.6	146.9	31.8	0	
FDCC12	151.6	349.8			323.1	146.9	47.7	4.6	
FDCC13	146.8	345.4			452.2	220.4	0	13.8	
FDCC14	149.2	347.6			387.6	220.4	15.9	9.2	
FDCC15	151.6	349.8			323.1	220.4	31.8	4.5	
FDCC16	154.0	352.0			258.4	220.4	47.7	0	

**Table 7 materials-16-01298-t007:** The orthogonal test results.

Test Number	Slump (mm)	Apparent Density (kg/m^3^)	Compressive Strength (MPa)	
			7 d	28 d
FDCC1	49	1730.8	24.18	30.75
FDCC2	56	1726.3	22.84	26.40
FDCC3	87	1716.2	19.59	23.77
FDCC4	99	1698.3	18.03	21.66
FDCC5	44	1765.6	24.47	29.60
FDCC6	51	1744.0	26.24	30.93
FDCC7	79	1747.3	23.61	26.58
FDCC8	84	1735.7	25.55	31.49
FDCC9	33	1849.5	28.03	32.72
FDCC10	41	1784.0	24.89	30.92
FDCC11	52	1767.3	20.42	27.89
FDCC12	61	1746.3	28.23	32.14
FDCC13	21	1849.6	25.39	31.59
FDCC14	29	1719.1	23.35	28.84
FDCC15	37	1704.4	22.41	27.49
FDCC16	60	1695.7	21.81	26.13

**Table 8 materials-16-01298-t008:** The orthogonal range analysis of various factors.

Performance Index	Range	Factor			
		*A*	*B*	*C*	*D*
Slump (mm)	*K_j_* _1_	72.75	36.75	53.00	54.50
	*K_j_* _2_	64.50	44.25	49.50	53.25
	*K_j_* _3_	46.75	63.75	58.25	58.00
	*K_j_* _4_	36.75	76.00	60.00	55.00
	*R* _1_	36.00	39.25	10.50	4.75
Apparent density (kg/m^3^)	*K_j_* _1_	1717.90	1798.88	1734.45	1735.88
	*K_j_* _2_	1748.15	1743.35	1735.65	1769.73
	*K_j_* _3_	1786.78	1733.80	1755.13	1740.38
	*K_j_* _4_	1742.20	1719.00	1769.80	1749.05
	*R* _2_	68.88	79.88	35.35	33.85
7-d compressive strength (MPa)	*K_j_* _1_	21.16	25.52	23.16	24.84
	*K_j_* _2_	24.97	24.33	24.49	23.55
	*K_j_* _3_	25.39	21.51	24.13	22.69
	*K_j_* _4_	23.24	23.41	22.98	23.68
	*R* _3_	4.23	4.01	1.51	2.15
	*K_j_* _1_	25.65	31.17	28.93	29.58
	*K_j_* _2_	29.65	29.27	28.91	29.34
	*K_j_* _3_	30.92	26.43	29.21	27.61
	*K_j_* _4_	28.51	27.86	27.69	28.20
	*R* _4_	5.27	4.73	1.52	1.97

**Table 9 materials-16-01298-t009:** The variance analysis.

Performance Index	Range	Factor			
		*A*	*B*	*C*	*D*
Slump (mm)	$F_{1}$	66.243 *****	79.367 *****	5.724 ***	$F_{0.01}$(3,3) = 29.46 $F_{0.05}$(3,3) = 9.28 $F_{0.1}$(3,3) = 5.39
Apparent Density (kg/m^3^)	$F_{2}$	3.611	5.400 *	1.272	
7-d Compressive Strength	$F_{3}$	4.742	3.660	0.687	
28-d Compressive Strength	$F_{4}$	5.773 *	4.660	0.522	

## Data Availability

All data are available in the main text.

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
