# Peer review of "The Workability and Mechanical Performance of Fly Ash Cenosphere–Desert Sand Ceramsite Concrete: An Experimental Study and Analysis"

_materials, 2023, doi:10.3390/ma16031298_

Round 1
Reviewer 1 Report
This work investigates the mechanical performance of fly ash cenosphere on desert sand ceramsite concrete. Although the author has used appropriate techniques for investigation, but the scientific and technical writing is very weak. For instance, the abstract is not reflecting the findings of the conclusion. Also, most of the sections are occupied with information while very little effort seems to be performed to present a deep related discussion that could convey the facts and findings.
On this stage, I would suggest that the author put more effort into analysing the data scientifically and importantly focus on technical writing and presentation as in the present form it is more like a conference paper.
The authors might find the following comments and corrections helpful in improving the present work.
· Abstract - Starts directly without mentioning the need / importance of conducting this study to highlight the research gap/ problem statement.
· Explain the significance of this research.
· Table 6 has a missing value of the amount of ceramsite raw material.
· Page 8 line 193 onwards - Please describe the statement properly or if using symbols, please represent it in equation manner to properly highlight the relation.
· Page 9 line 202 to 207 - Please use uniform formatting for the symbols in the text as used in the respective table. Please try to write in italics bold or simple italics
· Table 3, 4, 5, 8 caption - Repetition/Spelling - Please pay attention to the captions (Select relevant and appropriate words expressing significance of the content). Don't use 'full stop' at the end.
· Table 9 has no double starick symbol but the cross reference in line 206 explaining it being significant? Please recheck.
· Conclusion should be specific based on the outcome of the study rather than on generic statements. Please involve some numeric quantification to add more value.
· Finally, I strongly suggest amending the English language and correcting typos. The overall paper needs to be reassessed for major grammatical mistakes and improper writing practice.
Reviewer 2 Report
Please refer to the attachment.

Reviewer 3 Report
Well presented and written.
Author Response
Thank you very much for reviewing and approving our paper, and for your hard work in guiding us, which has been professionally touched up. Here are the relevant documents and the retouched article

Reviewer 4 Report
A careful review of the manuscript “Working and mechanical performance of fly ash cenosphere desert sand ceramiste concrete: test and analysis” has been completed. Despite the fact that the authors tried to develop an analysis method to evaluate the working and mechanical performance of fly ash cenosphere - desert sand ceramsite concrete, it is not really clear what is the main objective of the manuscript and how these materials can be implemented in practice; also, it is very difficult to follow what is presented. This paper is useful for engineers to see the effects of desert sand replacement rate, fly ash cenosphere replacement rate and polymer emulsion. However, this investigation is not comprehensive and there is still room to improve. Therefore, this manuscript is not recommended for publication in Journal of Materials due to the fact that paper has a critical and serious problem explained below:
1. English needs to be improved. I had difficulty following the text and had to read the same sentence several times.
2. The originality is not explained in detail.
3. References are not cited sufficiently and appropriately.
Round 2
Reviewer 2 Report
Thanks to the authors, the quality of the paper has been dramatically increased and the paper has been improved. It sounds satisfying to me.
Good Luck
Reviewer 4 Report
There are still room to improve!